# A Visually Inspired Computational Model for Recognition of Optic Flow

**Xiumin Li** [1,*], **Wanyan Lin** [1] , **Hao Yi** [2], **Lei Wang** [1] **and Jiawei Chen** [1]

1   College of Automation, Chongqing University, Chongqing 400030, China
2   Huawei Technologies Co., Ltd., Shenzhen 518129, China
*   Correspondence: xmli@cqu.edu.cn

**Abstract:** Foundation models trained on vast quantities of data have demonstrated impressive performance in capturing complex nonlinear relationships and accurately predicting neuronal responses. Due to the fact that deep learning neural networks depend on massive amounts of data samples and high energy consumption, foundation models based on spiking neural networks (SNNs) have the potential to significantly reduce calculation costs by training on neuromorphic hardware. In this paper, a visually inspired computational model composed of an SNN and echo state network (ESN) is proposed for the recognition of optic flow. The visually inspired SNN model serves as a foundation model that is trained using spike-timing-dependent plasticity (STDP) for extracting core features. The ESN model makes readout decisions for recognition tasks using the linear regression method. The results show that STDP can perform similar functions as non-negative matrix decomposition (NMF), i.e., generating sparse and linear superimposed readouts based on basis flow fields. Once the foundation model is fully trained from enough input samples, it can considerably reduce the training samples required for ESN readout learning. Our proposed SNN-based foundation model facilitates efficient and cost-effective task learning and could also be adapted to new stimuli that are not included in the training of the foundation model. Moreover, compared with the NMF algorithm, the foundation model trained using STDP does not need to be retrained during the testing procedure, contributing to a more efficient computational performance.

**Keywords:** foundation model; MT-MSTd; STDP; SNN; optic flow

**MSC:** 62M45

## 1. Introduction

Recently, foundation models based on deep artificial neural networks have shown robust representations of their modeling domain and achieved breakthroughs for accurately predicting neuronal responses to arbitrary natural images in a visual cortex [1–6]. However, despite the appearance of deep learning artificial neural networks, which have recently shown remarkable capability on a broad range of computational tasks, these models require high energy consumption and need to run on graphics processors that consume many kilowatts of power [7]. Therefore, brain-inspired algorithmic models and neuromorphic hardware processors are increasingly emerging and can potentially lead to low-power intelligent systems for large-scale real-world applications.

Spiking neural networks (SNNs) have attracted significant attention from researchers across various domains due to their brain-like information processing mechanism. However, training SNNs directly on neuromorphic hardware remains a significant challenge due to the non-differentiable nature of the spike-generation function. Amirhossein T. proposed a method called BP-STDP, which uses the difference between the desired output sequence and the actual firing sequence for backpropagation [8]. Converting the well-trained rate-based ANNs to SNNs by directly mapping the connection weights has also been broadly

studied [9,10]. Zhang et al. demonstrate that both the learning speed and the robustness of computation accuracy can be significantly improved by applying the biologically inspired intrinsic plasticity learning scheme into spiking feed-forward neural networks [11,12]. Kim S. from Seoul National University proposed Spiking-YOLO, which was the first application of SNNs in object detection, achieving a performance comparable to CNNs with extremely low power consumption [13]. However, these efforts to improve learning algorithms based on gradient descent iterations are far from the mechanism of real visual processing in the brain cortex, with the limitations of single network structure, high computational cost, and lack of biological plausibility.

It has been known that sparse coding, also known as sparse representation, uses a small number of elements to represent most or all of the original signals and explains how the visual cortex achieves efficient encoding and processing of information using a small number of neurons in an iterative manner. Therefore, modeling SNNs based on visually inspired information processing is of great significance for improving computational performance. In 1999, Professor Daniel D. proposed non-negative matrix factorization (NMF), which enables neural networks to achieve modular recognition of images and has significant implications for visual cognitive computational models [14]. Some evidence suggests that its dimensionality reduction and sparsity constraints are an effective form of population coding [15]. Beyeler M. proposed a hypothesis that the medial superior temporal region (MSTd) effectively encodes various visual flow patterns from neurons in the middle temporal region (MT) [16], and the sparse coding process is similar to NMF [17]. Further, an SNN model based on evolved and homomorphic synaptic scaling (STDP-H) learning rules was proposed and confirmed this hypothesis [18]. This model learns a compressed and efficient representation of input patterns similar to NMF, thus generating a receptive field similar to what is observed in the MSTd of monkeys. This suggests that the observed STDP-H in the neural system may have similar functionality to NMF with sparse constraints, providing an experimental platform for theoretical mechanisms on how MSTd efficiently encodes complex visual motion patterns to support robust self-motion perception.

Neurons in the visual cortex receive optic flow-like input and inherit their speed and direction preferences to process video sequences of realistic visual scenes. Optic flow can be used to avoid obstacles and approach goals in novel cluttered environments for animals by perceiving the heading direction of self-motion and guiding locomotion. Therefore, accurate recognition of optic flow is crucial for the development of visually based navigation models, such as the ViSTARS neural model, which was developed to describe neuronal information processing of the V1, MT, and MSTd areas in the primate visual dorsal pathway [19,20]. In [21], a biologically inspired neural network that learns patterns in optic flow is proposed based on fuzzy adaptive resonance theory. The paper [22] presented an artificial neural network that accurately estimates the parameters describing the observer's self-motion from MSTd-like optic flow templates. However, how to efficiently recognize the patterns of optic flow based on spiking neural networks and spike-based learning rules has not been fully studied yet.

Based on the above-mentioned literature, a visually inspired computational model inspired by the image processing mechanisms of the primary visual cortex for the recognition of optic flow is proposed in this paper. The visually inspired SNN model serves as a foundation model that is trained using spike-timing-dependent plasticity (STDP) on large amounts of optic flows for extracting core features. The ESN model trained using the linear regression method makes readout decisions for recognition tasks of optic flow. The results show that STDP can perform similar functions as non-negative matrix decomposition (NMF), i.e., generating sparse and linear superimposed readout based on local features, which is an ideal core/foundation model to accurately reconstruct input stimuli for image reconstruction. Based on the well-trained core network, ESN can accurately recognize not only optical flows but also new stimuli such as the grating stimulus. Moreover, ESN requires significantly less training data than the one without a fully trained core SNN

model. Compared with the NMF algorithm, the foundation model trained with STDP does not need to be retrained during the testing procedure for new stimuli, contributing to more efficient computational performance.

In summary, our main contributions are: (1) a foundation model for the recognition of optic flow is established based on biologically realistic spiking neural networks and trained with a neurophysiologically observed learning rule (STDP); (2) the efficiency and necessity of feature extraction from the foundation model is confirmed from our simulation results. That is, the computational performance mainly depends on feature extraction instead of readout training, indicating that learning is a multi-stage feature-extraction process instead of end-to-end training, as commonly used in neural networks.

## 2. Materials and Methods

The overall architecture of the model is depicted in Figure 1. There are two sub-networks: the foundational model of SNN and the decision model of ESN, which are trained separately. Since it is known that neurons in the MSTd region of the visual cortex can efficiently recognize firing patterns from neurons in the MT region, here the SNN model consists of two visual cortical layers, i.e., the MT layer and the MSTd layer. The input optic flow stimuli are initially processed by a group of MT neurons, which encode the information into Poisson spike trains. These spike trains are then transmitted to a group of excitatory spiking neurons representing MSTd. The connection weights from MT to MSTd are updated using spike-timing-dependent plasticity (STDP) based on large amounts of optic flow with different patterns to extract the core features of the training dataset. After training, all core features are saved in the form of MT-MSTd synaptic weights. Then, the firing rate of the MSTd layer encoding the relative importance of the core's features to the current visual input is transformed into an ESN model trained using the linear regression method, which makes readout decisions to recognize which patterns the input stimuli belong to. Detailed models are described in the below subsections.

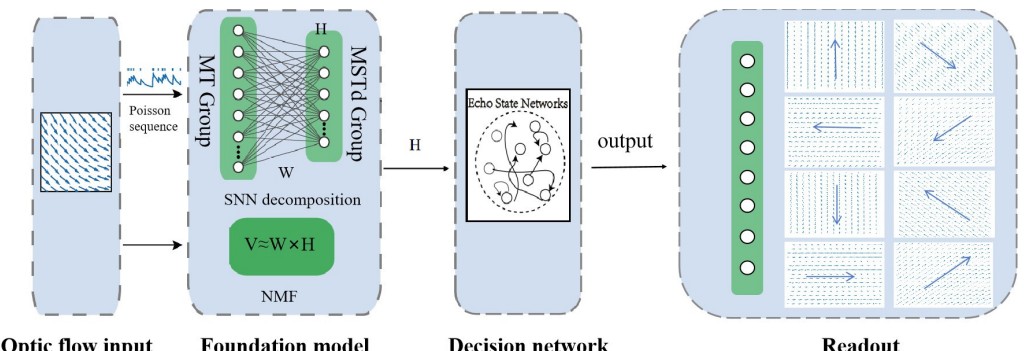

**Figure 1.** Model architecture. This model consists of three aspects: visual input, foundation model of the visual cortex, and decision network. The input stimuli to the network are computer-generated $15 \times 15$ pixel arrays representing optic flow. Inputs encode information into Poisson spike trains, which are then transmitted to the MT layer. The architecture of the MT group is $15 \times 15 \times 8$, that of the MSTd group is $8 \times 8 \times 1$, and the Inh group is $8 \times 8 \times 8$. Both MSTd-to-Inh and Inh-to-MSTd connections follow uniform random connectivity with a 0.1 probability. STDP learning rule is employed to adjust the network weights from MT to MSTd. ESN consisting of a random reservoir network and readout layer is trained for making decisions for recognition tasks.

### 2.1. Visual Input

The input stimuli to the network are computer-generated $15 \times 15$ pixel arrays representing optic flow. To simulate the visual motion on the retina caused by an observer moving in a 3D environment, a motion field model is utilized to generate the optic flow stimuli. Please refer to the papers [17,23] for details. Using computer-generated data, we created 6000 samples of optic flow data. Each sample contains input information for

different directions and velocities, where the values represent the length of the optic flow vectors. Therefore, when plotting, the information of the 8 directions and velocity is superimposed within a $15 \times 15$ grid. This means that the vectors are converted into values on the $x$ and $y$ coordinates and added together to obtain the coordinates of the vector endpoints. The MATLAB function "quiver" can be used to plot the optic flow vector map. Figure 2 depicts optic flow maps with different patterns that simulate the projection of three-dimensional coordinates onto the retina as the observer moves in a three-dimensional world. From the figures, it can be observed how the projection of the three-dimensional points on the retina changes when the observer moves backward, rotates, and translates in the three-dimensional world. Input stimuli encode information into Poisson spike trains, which are then transmitted to a group of excitatory spiking neurons in the MT layer.

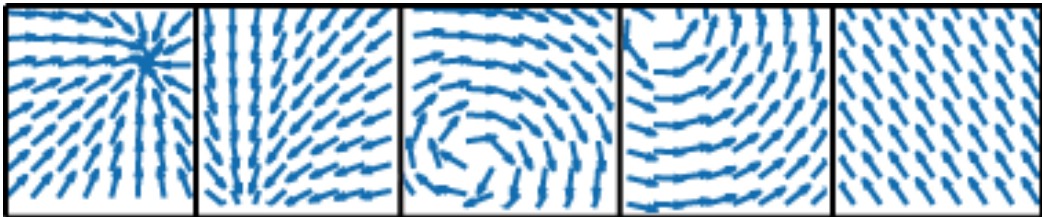

**Figure 2.** Sample plots of visual optic flow to simulate the projection of three-dimensional coordinates on the retina as the observer moves in the three-dimensional world [17].

### 2.2. Foundation Model of Spiking Neural Networks

This SNN model focuses on extracting core features of optic flow patterns resulting from self-movement. Each optic flow field is processed by MT units that resemble the orientation selectivity of MT neurons, which selectively respond to optic flows with specific positions, directions, and velocities. Here, the MT layer is composed of $15 \times 15 \times 8 = 1800$ units, representing the responses to the specified eight directions ($45°$, $90°$, $135°, 180°$, $225°$, $270°$, $315°$, $360°$) and one velocity. The population code of local direction and speed of motion acts as the activity pattern of these 8 units/pixel. All neurons within the MT layer are isolated and the neural activity is modeled using Izhikevich neurons [24] exhibiting excitatory regular spiking behavior. The input optic flow stimuli are initially processed by a group of MT neurons, which encode the information into Poisson spike trains. The average pixel value of input determines the mean firing rate of spike trains. These spike trains are then transmitted to the MSTd layer. The MSTd group contains 64 isolated Izhikevich neurons corresponding to 64 core features. The weight connections from MT to MSTd follow a Gaussian distribution, in which close neurons have a higher probability of connectivity and higher initial connection weights. Additionally, the MSTd group projects to a set of inhibitory neurons (Inh), providing feedback inhibition to regulate network activity. The inhibitory group has 512 neurons. The population size of MT is $15 \times 15 \times 8$, that of MST is $8 \times 8 \times 1$, and that of the inhibitory group is $8 \times 8 \times 8$. Both MSTd-to-Inh and Inh-to-MSTd connections follow uniform random connectivity with a 0.1 probability [25]. The detailed models for the Izhikevich neuron and synapses are as follows:

$$\frac{dV(t)}{dt} = 0.04V(t)^2 + 5V(t) + 140 - U(t) + I^{syn}(t) \tag{1}$$

$$\frac{dU(t)}{dt} = a(bV(t) - U(t)) \tag{2}$$

$$\text{if } V > V_{cutoff} \text{, then} \begin{cases} V = c \\ U = U + d \end{cases} \tag{3}$$

where $V$ and $U$ represent membrane potential and membrane recovery variables, respectively. $I^{syn}$ is the external synaptic current, the membrane potential $V$ has an $mv$ scale, and time has a $ms$ scale. The parameters $a, b, c, d$ in (2) and (3) are set as $a = 0.02, b = 0.2,$

$c = -65, d = 8$ or $a = 0.1, b = 0.2, c = -65, d = 2$, representing regular-spiking (RS) neurons (excitatory neurons) or fast-spiking (FS) neurons (inhibitory neurons). Here, we use the conductance-based description for synaptic models, which calculates the synaptic current using complex conductance equations for each synaptic receptor type. AMPA and NMDA are excitatory synaptic connection types. $GABA_A$ and $GABA_B$ are inhibitory synaptic connection types.

$$i_e = i_{NMDA} + i_{AMPA}$$
$$i_i = i_{GABA_A} + i_{GABA_B} \tag{4}$$

The total current $I_{syn}$ can be expanded to:

$$i_{syn} = -g_{AMPA}(v - v^{rev}_{AMPA}) - g_{NMDA} \frac{\left[\frac{v+80}{60}\right]^2}{1 + \left[\frac{v+80}{60}\right]^2}(v - v^{rev}_{NMDA})$$
$$- g_{GABA_a}(v - v^{rev}_{GABA_A}) - g_{GABA_B}(v - v^{rev}_{GABA_B}) \tag{5}$$

where $g$ and $v_{rev}$ are specific to a particular ion channel or receptor. The synaptic conductance $g$ obeys the exponential decay and changes when presynaptic spikes arrive.

$$\frac{dg_r(t)}{dt} = -\frac{1}{\tau_r} g_r(t) + w \sum_i \delta(t - t_i) \tag{6}$$

where $\delta$ is the Dirac delta function, and $r$ is the receptor type ($AMPA, NMDA, GABA_a, GABA_b$). $t_i$ is the presynaptic spikes arrival time [24].

Since topological structure enhances temporal-spatial processing ability and biological plausibility, the training of the synaptic matrix $W$ from MT to MSTd directly affects the effectiveness of feature extraction of the foundation model. In this paper, all connections in SNN are plastic, whose weight values are modulated by the heterosynaptic STDP (STDP-H) to optimize the learning rule parameters based on an evolutionary algorithm, as proposed in [18]. Simulations were conducted using the CARLSim SNN simulator platform https://github.com/UCI-CARL/CARLsim6 (accessed on 1 March 2022) [26,27].

The model of STDP-H can be described as follows [18]:

$$\frac{dw_{i,j}}{d_t} = \left[\overbrace{\alpha \cdot w_{i,j}(1 - \overline{R}/R_{t\,arg\,et})}^{\text{hom}eostasis} + \overbrace{\beta(LTP_{i,j} + LTD_{i,j})}^{STDP}\right] \cdot K \tag{7}$$

Equation (7) describes the cumulative impact of STDP-H on a specific synapse $w_{i,j}$ that connects the presynaptic neuron $i$ and the postsynaptic neuron $j$. Equation (7) consists of two key components. The first component pertains to homeostatic scaling, which is determined by the ratio of the average firing rate $R$ to the target firing rate $R_{t\,arg\,et}$ of neuron $j$. Homeostatic scaling modulates the rate of synaptic weight changes, reducing it when the neuron is excessively active and increasing it when the neuron is too inactive. The second component in Equation (7) deals with spike-timing-dependent plasticity (STDP [28], encompassing both long-term potentiation (LTP) and long-term depression (LTD). STDP adjusts the strength of synaptic connections based on the timing of spikes between presynaptic and postsynaptic neurons. Specific details can be found in reference [18].

For STDP learning, parameter optimization was performed with the Parameter Tuning Interface in CARLsim 6, which used the Evolutionary Computations in JAVA library (ECJ) [29]. By leveraging the parallel execution capabilities provided by CARLsim, the computations were distributed among GPU and CPU cores, resulting in a significant acceleration of the simulation process. ECJ was used to evolve suitable parameters for STDP learning. It automatically constructs multiple independent individuals based on the network structure and selects the best-adapted parameters according to evaluation rules. During learning, the input data were shuffled. A portion of the samples was used for training to adjust the MT-to-MST connection weights. The remaining samples were used for testing and parameter evaluation. Each sample ran for 0.5 s before stopping the Poisson

process for spike generation. An additional 0.5 s of idle time was added between samples to allow neuronal voltages to decay without affecting subsequent inputs. The evaluation metric was the correlation coefficient between the input and reconstructed samples.

$$fitness = \frac{\sum\limits_{m}\sum\limits_{n}(A_{mn} - \bar{A})(B_{mn} - \bar{B})}{\sqrt{\sum\limits_{m}\sum\limits_{n}(A_{mn} - \bar{A})\sum\limits_{m}\sum\limits_{n}(B_{mn} - \bar{B})}} \tag{8}$$

The correlation coefficient is calculated as: $\bar{A}$ and $\bar{B}$ are the column-wise mean values of matrices $A$ and $B$, respectively. $A$ is the test samples from the input data, and $B$ is the product of the MT-to-MST connection weight matrix $W$ and the MST neuronal firing rate matrix $H$. The ECJ parameters were configured with adjustment ranges as mentioned in [25]. Each iteration evaluated 15 network individuals, and after 100 iterations the best fitness of 72.66% was achieved. The 18 parameters corresponding to the highest fitness network were selected as the adapted parameters.

### 2.3. Decision Model of Echo State Network

Based on the above well-trained foundation SNN model, all core features are saved in the form of MT-MSTd synaptic weights. Then, the firing rates of MSTd neurons encoding the relative importance of the core features to the current visual input are transferred to the readout decision model. In this paper, an echo state network (ESN) proposed by Jaeger [30–32] is utilized as the decision model. It consists of a random sparse network (reservoir) and one readout layer. The reservoir acts as an information processing medium, which maps the low-dimensional input signal to the high-dimensional state space.

Specifically, we have the following weight matrices: $W_{in}$, $W_{res}$, $W_{back}$, and $W_{out}$. $W_{in}$ signifies the weight matrix governing connections from the input layer to the reservoir, $W_{res}$ represents the internal weight matrix regulating connections within the reservoir, and $W_{back}$ is responsible for feedback connections from the output layer to the reservoir. Lastly, $W_{out}$ corresponds to the weight matrix that manages connections from the reservoir to the output layer. In terms of variables, we use $u(n)$ to denote the network input at time $n$, $x(n)$ signifies the state vector representing the network's reservoir, and $y(n)$ represents the network's output.

The updated reservoir status is calculated according to Formula (9) and the network output is calculated according to the following Formula (10) :

$$x(n+1) = f(W_{in}u(n+1) + W_{res}x(n) + W_{back}y(n)) \tag{9}$$

$$y(n+1) = f_{out}(W_{out}[x(n+1)|u(n+1)] \tag{10}$$

where $f(\bullet)$ is the activation function of the neuron of the network reservoir. $f(\bullet)$ can be typically defined as a sigmoid or tanh function [33], and $fout(\bullet)$ represents the output function of the output layer neurons. In ESN, $W_{in}$, $W_{res}$, and $W_{back}$ are randomly generated before training and no longer adjusted. $W_{out}$ is the only matrix that needs to be learned. Algorithms such as the ridge regression algorithm are used to learn $W_{out}$.

There are 500 units in the reservoir network. Only synaptic weights from the readout layer to the reservoir are updated using the linear regression method and connections within the reservoir are randomly generated. Furthermore, the connection weights in the foundation model of SNN are also frozen and unchanged during the readout training process. This ESN model for recognition tasks makes readout decisions to recognize which patterns of the input stimuli belong. The ESN network structure parameters for this experiment are shown in Table 1.

**Table 1.** Key parameters of ESN model.

| ESN Parameters | Values |
|---|---|
| Reservoir Sparsity (SP) | 0.5 |
| Displacement Scale (IS) | 1 |
| Input Unit Scale (IC) | 1 |
| Spectral Radius (SR) | 0.85 |
| Reservoir Activation Function (f) | *Tanh* |
| Output Unit Activation Function ($f_{out}$) | 1 |
| Regularization Coefficient ($\lambda$) | $1 \times 10^{-3}$ |

## 3. Results

The proposed model framework, as shown in Figure 1, consists of three aspects: input layer, feature extraction in the foundation model of the visual cortex, and decision layer. First, optic flow data $V$ are encoded as spike sequences using Poisson frequency encoding, serving as the input to the MT-group neurons in the SNN model. Synaptic weights from MT to MSTd are saved in the $W$ matrix, and the firing frequency of the MSTd group is represented as the $H$ matrix. The STDP learning rule is employed to adjust the network parameters, aiming to reconstruct the original optic flow data $V$ with the $W \times H$ matrix and automatically optimize the network by evaluating the reconstruction performance at each iteration. Finally, the effectiveness of the foundation SNN model is validated through image recognition of the optic flow and grating stimulus by training the ESN model. The pseudo-code is as follows (Algorithm 1):

---

**Algorithm 1:** A visually inspired computational model for recognition of optic flow

---

1 **Step 1: Foundation model of SNN**
2 **for** *each optic flow data* **do**
3      Encode input optic flow into Poisson spike trains
4      Adjust MT-MSTd weight connections $W$ with STDP-H learning
5      Compute MSTd firing rates $H$
6      Reconstruct input data $V$ to verify the efficiency of feature extraction using
       $V \approx W \times H$
7 **Step 2: Decision model of ESN**
8 **for** *each $H_i$* **do**
9      Initialize synaptic weights randomly
10      Use ridge regression algorithm to obtain $W_{out}$ while keeping the other weights
       fixed
11      Achieve recognition for optic flow

---

### 3.1. Feature Extraction of SNN Model

Non-negative matrix factorization (NMF) is known for its key feature of feature extraction by decomposing data into linear combinations of different local features [14,34]. This algorithm is a sparse decomposition processing with dimensional reduction, which has been previously shown to be capable of mimicking a wide range of monkey MSTd visual response properties [15,17]. The input stimuli can be accurately reconstructed from a linear superposition of the sparse, parts-based features regarded as basis flow fields (see Figure 3a). Considering the columns of an input matrix $V$ with a set of samples, these data can be linearly decomposed as $V \approx W \times H$, where the columns of the matrix $W$ contain the basis feature vectors of the decomposition and the rows of $H$ contain the corresponding coefficients that give the contribution of each basis vector. In NMF, the basis functions $W$ and the corresponding coefficient vector $H$ are obtained simultaneously in one calculation.

Although the basis matrix $W$ obtained in NMF can be considered the synaptic weights of a population of simulated neurons in a neural network, both the $W$ and $H$ matrices need to be retrained every time a new sample is added to the input matrix $V$, that is, all of the data must be re-decomposed.

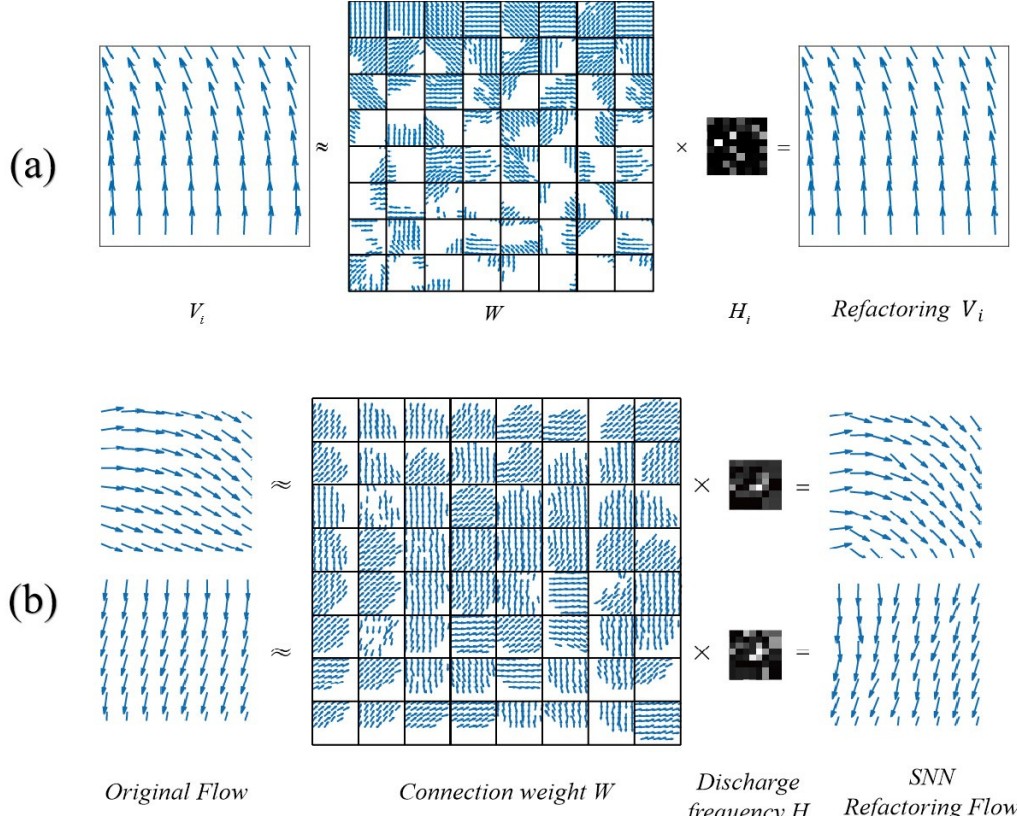

**Figure 3.** Decomposition and reconstruction of optic flow from NMF (**a**) and STDP learning (**b**). $V_i$ is a single sample. Basis flow fields are shown in an $8 \times 8$ MT-MSTd weight matrix $W$, which is visualized using the population vector decoding method. $H$ is a grayscale image containing weight information of core features. Input optic flow $V$ can be reconstructed from a sparse activation of model MSTd neurons, i.e., $V \approx W \times H$. The optic flow of each square in the basis matrix $W$ is different, indicating that the features are separated and contain the local features of the original data. By multiplying the coefficient matrix $H$, the features in $W$ are combined to reconstruct the original data. Therefore, the coefficient matrix $H$ contains the weight of each local feature and can represent the original data for classification verification.

Recent studies have shown that the SNN model can learn efficient representations and reconstructions of the input patterns similar to that emerging from NMF [18]. Therefore, we adopt SNN with STDP learning to achieve feature extraction of optic flow. Figure 3b demonstrates that a set of basis flow fields (shown in an $8 \times 8$ MT-MSTd weight matrix $W$, which is visualized as basis flow fields using the population vector decoding method) can emerge from both NMF and STDP learning, indicating that the core features are successfully extracted from the original input data. Meanwhile, optic flow fields can be reconstructed from a sparse firing activation of model MSTd neurons that prefer various orientations of the basis flow fields. That is, the optic flow input samples $V$ can then be approximated by a linear superposition of basis flow fields, i.e., $V \approx W \times H$, which means that the original 1800-dimensional input data $V$ are compressed into a 64-dimension coefficient matrix H. In this way, sparse coding reduces the overall neural activity necessary to represent the information. This not only reduces the number of nodes in the MSTd layer but also improves the learning efficiency of the decision layer.

Unlike the NMF method, once the connection MT-MSTd weight matrix *W* is well-trained, it can be frozen and the parts-based features saved in *W* can be re-used or shared for new input stimuli. The coefficient vector *H* that gives the contribution of each basis feature for the current input stimuli can be directly obtained through the firing responses of the MSTd layer. Inputs belonging to the same patterns share similar value distributions of the *H* vector, while *H* coefficients are different for different input patterns. To validate the effectiveness of this feature-extraction method, an ESN model is used to train the readout decision layer for the recognition task of optic flow.

### 3.2. Recognition Performance

Image reconstruction based on feature extraction is shown in Figure 3b, which indicates that good image reconstruction means accurate feature representation of the MSTd neurons. In order to test the effectiveness of the SNN model, eight categories with different directions ranging from 0° to 360° in 45° increments are selected from a sample of 6000 MT optic streams that had previously been generated. As shown in Figure 1, the readout neurons are divided into eight classes corresponding to the superimposed directions. Each class included 120 samples, resulting in a total of 960 samples. A fully trained foundation model has the capability to achieve more diverse parts-based features than the partially trained foundation model (see Figure 4a). The weight visualization of a fully trained model demonstrates more distinct and concrete feature extraction compared to the incompletely trained model. Therefore, we can see from Figure 4b that the fully trained base model significantly outperforms the incompletely trained counterpart, contributing to the reduction in training samples for readout learning.

Furthermore, we conducted a comparative analysis of the classification performance between the fully trained SNN model and a basic CNN model, as depicted in Figure 4c. The CNN architecture employed in this paper consists of six layers. The initial layer is the input layer, which receives data of size $15 \times 15 \times 8$. Following this is the second layer, a convolutional layer comprising 16 convolutional kernels of size $3 \times 3$ utilized to extract features from the input. Subsequently, the third layer is the activation function, employing the rectified linear unit (ReLU) activation function to introduce non-linear characteristics and enhance the network's expressive capability. The fourth layer is a pooling layer, utilizing a $2 \times 2$ max-pooling operation to reduce the dimensions of the feature maps. Next is the fully connected output layer containing ten nodes. Finally, through the combination of a Softmax layer and a classification layer, the network completes its output and transforms it into a probability distribution for executing the ultimate classification task. Since CNN is trained end-to-end for all layers, pattern recognition is examined with different numbers of training data.

Remarkably, we observed that our two-layer SNN model with ESN readout outperforms the six-layer CNN throughout the whole experimental process. Note that even when inputting different pattern samples, retraining for SNN is unnecessary, i.e., the weights in the SNN model are frozen and only the ESN readout is trained to learn new patterns. In contrast, the CNN model shows lower classification accuracy, and all layers need to be end-to-end retrained when new input patterns are added. This result demonstrates the advantages of our model in terms of low power consumption and high precision of computation.

Our model not only excels in recognizing visual optic flow (in-domain) but also accurately recognizes out-of-domain stimuli, such as the sinusoidal grating stimulus [35]. We use the model of the V1 visual cortex proposed in [36] to transfer the stimuli into the optic flow, as shown in Figure 5a. The left image represents the input grating stimulus, where *v* represents the direction of the grating motion. The right image represents the integrated optic flow. The results demonstrate that the V1 response accurately identifies the motion direction of the grating stimulus, indicating that the grating stimulus is transformed into optic flow stimuli and can be used for recognition validation of this model.

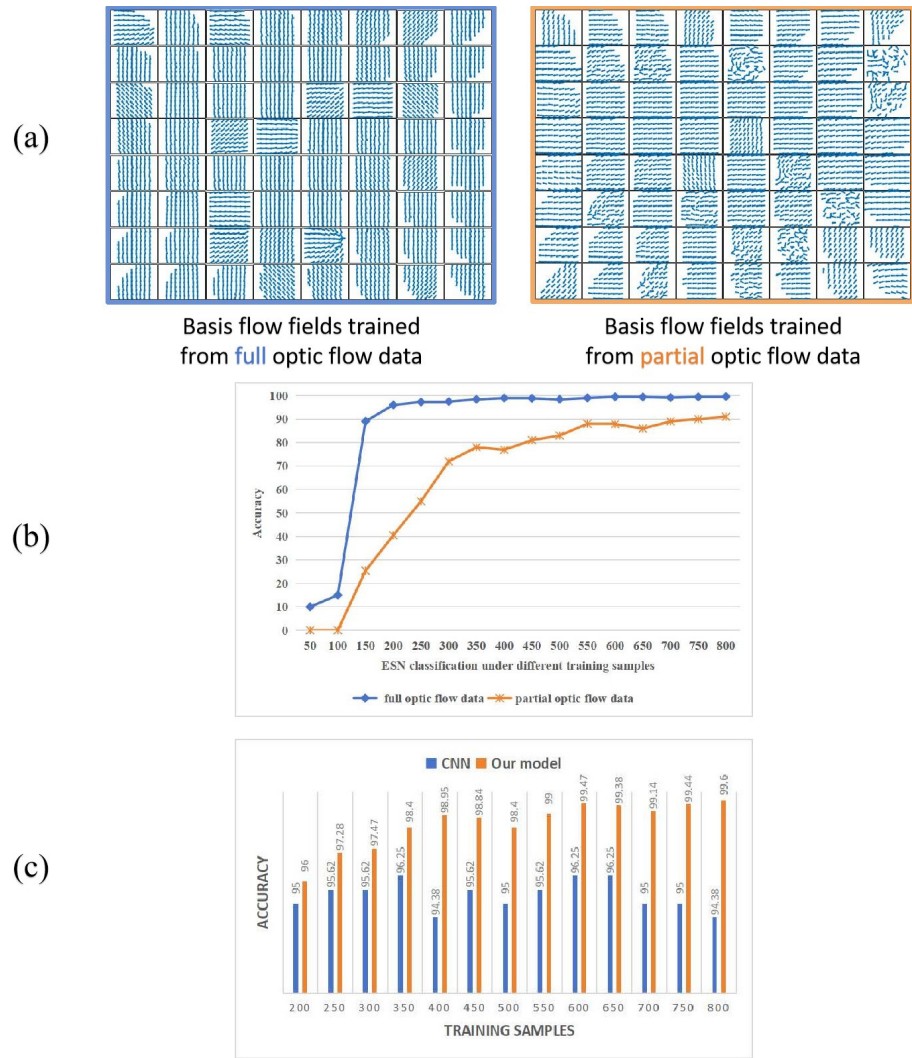

**Figure 4.** Fully trained foundation model can considerably reduce the training samples required for recognition tasks during the readout learning. (**a**) Basis flow fields trained from full or partial input data. (**b**) Performance comparison of fully trained or not fully trained foundation models under different training samples for readout learning. (**c**) Comparison of classification performance between a six-layer CNN model and our proposed model on optic flow data.

Based on the above-obtained foundation model trained from the optic flow stimulus, here the MT-MSTd connection weights in SNN are fixed. Training data from the new stimulus are only used to fit the readout weights of the ESN model. The recognition performance for the grating input is shown in Figure 5b. It can be seen that the training accuracy for the grating stimulus can reach up to 99.85% and the testing accuracy is 99.25%. This indicates that the foundation SNN model performs exceptionally well in classifying new stimulus data with no need to retrain the basis features, even when compared to its performance in recognizing optic flow data. This result reveals that a foundation model trained from vast quantities of data has remarkable capabilities and generalization in performing computational tasks.

However, if the grating stimulus is directed and projected to the SNN model without the processing of the V1 layer, recognition will not succeed. Because the MT layer can only process orientation-based input, images must be processed with V1 before input to the MT layer. Preliminary feature extraction for stripe shapes with V1 is also crucial for image recognition. Therefore, further study considering the V1 and V2 visual cortex as the foundation model is necessary for expanding its applicability to the recognition of broader image datasets.

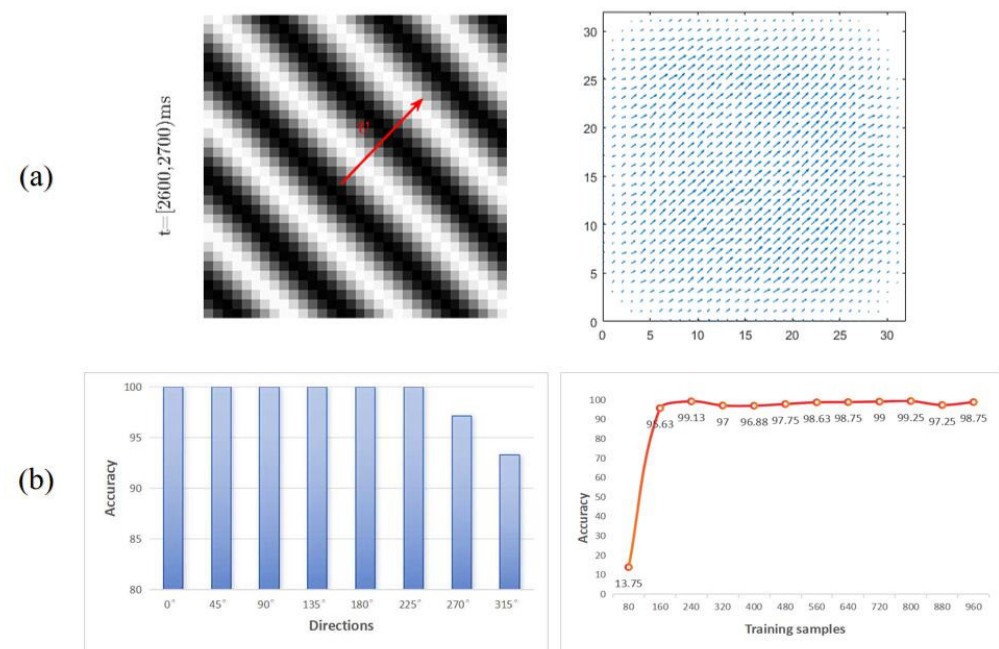

**Figure 5.** Recognition performance for grating stimulus. (**a**) **Left**: grating motion stimulus sample. **Right**: optic flow transformed from the original grating stimuli using V1 visual cortex model proposed in [36]. (**b**) **Left**: Average classification accuracy in each direction. **Right**: Classification accuracy with different numbers of grating training samples.

## 4. Conclusions

We introduce a visually inspired computational model that achieves efficient performance in the recognition of optic flow. Here, a foundation model based on SNNs with STDP learning is shown to be an efficient sparse coding for parts-based representation, where any flow field could be represented by only a small set of simulated MSTd neurons ($H$ coefficient) as compared with vast quantities of input samples. The results show that STDP can perform similar functions as NMF, i.e., generating sparse and linear superimposed readouts based on basis flow fields. The visually inspired SNN is an ideal core/foundation model to accurately reconstruct input stimuli for image reconstruction. Based on the well-trained SNN, the readouts can accurately recognize not only optic flow but also new stimuli such as the grating stimulus. Moreover, the ESN requires significantly less training data than the one without a fully trained core SNN model.

Our model based on biologically realistic SNNs offers a powerful new approach for the efficient recognition of visual optic flow, which has the potential to be used in neuromorphic applications to reduce computations. This study may give insight into the development of visually based navigation models, brain-inspired robots, and new-generation artificial intelligence machines.

**Author Contributions:** Methodology, W.L.; Formal analysis, J.C.; Resources, L.W.; Data curation, H.Y.; Supervision and Writing editing, X.L. All authors have read and agreed to the published version of the manuscript.

**Funding:** This paper is supported by STI 2030-Major Project 2021ZD0201300.

**Data Availability Statement:** The Evolutionary Computations in JAVA library (ECJ) used to evolve suitable parameters for STDP learning is available on https://cs.gmu.edu/eclab/projects/ecj/. The sinusoidal grating stimulus was generated from the Visual Stimulus Toolbox (https://zenodo.org/records/154061 for details).

**Acknowledgments:** The authors would like to give sincere appreciation to Jeffrey Krichmar and Kexin Chen for their insightful suggestions and discussions for this work.

**Conflicts of Interest:** Hao Yi was employed by the Huawei Technologies Co., Ltd. The remaining authors declare that the research was conducted in the absence of any commercial or financial relationships that could be construed as a potential conflict of interest

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
