# Peer review of "A Visually Inspired Computational Model for Recognition of Optic Flow"

_mathematics, doi:10.3390/math11234777_

Round 1

Reviewer 1 Report

Comments and Suggestions for Authors

The present manuscript addresses the issue of recognition of optic flow, using a SNN (with STDP learning) plus ESN model.

Authors claim their model performs similarly to a NMF, however no numerical comparison is provided. 

I strongly suggest they give an explicit comparison to support their claim. Alternatively they could give numerical comparisons with CNN or other models used to equivalent patter recognition, specially concerning their Fully vs Partial data statement.

The section Introduction is correct and the Methods are well explained.

However, section for Results could be improved, if a table for the accuracy with different hiperparameters (for the SNN and ESN) is presented.

Also, the Confusion matrix in Figure (5b) doesn't give any relevant information (almost all cells are empty) and could be perfectly excluded. 

Comments on the Quality of English Language

I have found minor spelling error - for instance, in Conclusions it appear a double "of of" which must be revised.

Reviewer 2 Report

Comments and Suggestions for Authors

This paper introduces a computational model, influenced by the image processing mechanisms of the primary visual cortex, designed to recognize optic flow. The visually-inspired Spiking Neural Network (SNN) model is the foundational component, trained through spike-timing-dependent plasticity (STDP) on extensive optic flow data to extract essential features. The Echo State Network (ESN) model, trained using linear regression, is employed for making readout decisions in optic flow recognition tasks. The results demonstrate that STDP can emulate the functionalities of non-negative matrix decomposition (NMF), producing sparse and linearly superimposed readouts based on local features. This suggests that STDP is an optimal foundational model for accurately reconstructing input stimuli in image reconstruction. This work is exciting and can be helpful for many applications. After reading the manuscript, here are my comments:

  1. Please clarify the novelty and contribution of this work. How does the approach proposed by the authors differ from similar approaches in identifying optic flow? Authors claim that the effective identification of optic flow patterns through spiking neural networks remains an area that has yet to be comprehensively explored. However, there is much research on this issue. Please clarify why this approach is so crucial. 
  2. Examining the proposed model, comprising the foundational aspect of Spiking Neural Networks (SNN) and the decision-making component of Echo State Networks (ESN), is necessary for comparison with other computational models. Additionally, a discussion about the results is crucial to identify new findings.
  3. Kindly elucidate the reasons behind the efficiency demonstrated by the proposed model in recognizing optic flow, particularly noting that it achieves a training accuracy of up to 99.85% and a testing accuracy of 99.25% for grating stimuli.
  4. There are many misformatted structures in the manuscript, such as the ref. in "....Amirhossein T. proposed a method called BP-STDP [8],...", "....the connection weights has also been broadly studied citeConversion of analog....", the caption details of Figure 1, all figures are not precise, etc.

I consider that the authors need to address these issues before their paper can be published in the journal.

Comments on the Quality of English Language

Some minor editing of the English language is required.

Round 2

Reviewer 1 Report

Comments and Suggestions for Authors

Authors have fulfilled the required changes.

Author Response

Thank you for your helpful suggestion and comments.

Reviewer 2 Report

Comments and Suggestions for Authors

In this version, the authors have added some experimental results and modifications to respond positively to my questions.

Comments on the Quality of English Language

Just minor editing of the English language is required.

Author Response

Thank you for your suggestion. The minor editing of the English language is improved in the revised manuscript.